# High–Normal Preoperative Potassium Level Is Associated with Reduced 30–Day Morbidity and Shorter Hospital Stay after Radical Cystectomy

**DOI:** 10.3390/jcm11051174

**Published:** 2022-02-22

**Authors:** Hendrik Borgmann, Mohamed M. Kamal, Anna Metzger, Robert Dotzauer, Nikita Fischer, Peter Sparwasser, Wolfgang Jäger, Igor Tsaur, Axel Haferkamp, Thomas Höfner

**Affiliations:** 1Department of Urology, University Medical Center, Johannes Gutenberg University, 55131 Mainz, Germany; hendrik.borgmann@unimedizin-mainz.de (H.B.); mohamedkamal.gheith@unimedizin-mainz.de (M.M.K.); robert.dotzauer@unimedizin-mainz.de (R.D.); fischer.nikita@unimedizin-mainz.de (N.F.); wolfgang.jaeger@unimedizin-mainz.de (W.J.); igor.tsaur@unimedizin-mainz.de (I.T.); axel.haferkamp@unimedizin-mainz.de (A.H.); thomas.hoefner@unimedizin-mainz.de (T.H.); 2University Hospital Frankfurt, 60590 Frankfurt, Germany; anna.metzger@web.de

**Keywords:** electrolytes, bladder cancer, urothelial carcinoma, cystectomy, potassium

## Abstract

Background: Radical cystectomy has high complication rates and, consequently, a high socioeconomic burden. The association between preoperative electrolyte levels and postoperative outcomes after radical cystectomy has not been investigated. Therefore, we aimed to investigate the association between preoperative potassium level and clinical (30-day morbidity) and economical (length of hospital stay) postoperative outcomes of patients undergoing radical cystectomy. Materials and Methods: We retrospectively evaluated clinical data of 317 patients who had undergone radical cystectomy for bladder cancer. Univariate and multivariate logistic regression analyses were performed to determine whether preoperative patient clinical factors influence clinical (30-day morbidity according to the Clavien-Dindo classification) and economical (length of hospital stay) postoperative outcomes. Results: In univariate analysis, low Charlson comorbidity score (*p* = 0.011), low ASA score *(p* = 0.015), no aspirin intake (*p* = 0.048) and high-normal preoperative potassium level (*p* = 0.034) were associated with reduced 30-day morbidity. In multivariate analysis, only high preoperative potassium remained an independent predictive factor for a reduced risk of postoperative complications (odds ratio 0.67, 95% confidence interval (0.48, 0.92), *p* = 0.014). Furthermore, high-normal preoperative potassium was the only preoperative factor associated with a shorter hospital stay ≤21 days (*p* = 0.007). Conclusions: High-normal preoperative potassium level was associated with better clinical (lower 30-day morbidity) and economical (shorter hospital stay) outcomes in patients undergoing radical cystectomy. We recommend that a randomized controlled trial be performed to investigate whether there is a causal relationship between preoperative potassium supplementation and postoperative complications and length of hospital stay.

## 1. Introduction

Radical cystectomy (RC), the gold standard treatment for muscle–invasive bladder cancer, has high complication rates of up to 57% major complications [1] and thus represents a high socioeconomic burden [2]. One of the most common complications of RC with urinary diversion is paralytic ileus, which causes a prolonged hospital stay and delayed recovery [3]. The first bowel movement marks a breakthrough point in the recovery process, highlighting the need to identify factors associated with accelerated bowel motility.

Protocols to enhance recovery after RC surgery for bladder cancer have been implemented [4]. These include preoperative loading with high–calorie and electrolyte-enriched drinks [5]. Moreover, a recent randomized trial of patients undergoing RC reported accelerated recovery of normal gastrointestinal function after perioperative administration of a potassium-enriched, chloride-depleted 5% glucose solution [6]. However, despite these efforts to optimize electrolyte levels in the perioperative management of RC, the association between preoperative electrolyte levels and postoperative outcomes after RC has not yet been explored.

Therefore, we aimed to investigate the association of a large panel of preoperative variables (patient characteristics, vital signs and blood values), with a special focus on preoperative potassium levels, with clinical (complications) and economical (length of hospital stay) postoperative outcomes after RC.

## 2. Methods

### 2.1. Data Collection

After institutional review board approval (reference number 32/16), we collected the clinical and histopathological data of patients who had undergone RC for bladder cancer between 2002 and 2015 at University Hospital Frankfurt. All surgeries were performed by experienced surgeons with a minimum caseload of 30 surgeries for RC. The type of urinary diversion used was most commonly ileal neobladder for continent diversion and ileal conduit for incontinent diversion. Preoperative, intraoperative and postoperative outcome variables were registered in a database. We focused on identifying the preoperative variables that are associated with clinical (complications) and economical (length of hospital stay) postoperative outcomes. Therefore, we included comprehensive preoperative patient data including comorbidities (Charlson comorbidity score, American Society of Anesthesiologists (ASA) score), medications, vital signs (blood pressure and heart rate), and a set of nine preoperative blood values: hemoglobin, leucocytes, thrombocytes, C-reactive protein, creatinine, MDRD, glucose, sodium, and potassium. The study endpoints were 30-day morbidity and length of hospital stay. Complications that occurred within 30 days after surgery were documented and classified according to the Clavien-Dindo classification system. We registered the length of hospital stay in days from admission until discharge.

### 2.2. Statistical Analysis

The frequencies and proportions are presented for categorical variables and means and standard deviations for continuously coded variables that were normally distributed. For our main analyses, we defined binary (favorable and unfavorable) outcome groups for each of the three outcome measurements: number of complications (low vs. high), presence of a major complication (yes vs. no), and hospital stay (short vs. long). We calculated cut-off values to create similar group sizes for these outcome measurements. To allow for statistical testing of multiple variables for a limited number of events, we distributed patients into up to three groups for each preoperative variable. The effect of different potassium levels in patient subgroups had been reported before with regard to cardiovascular outcomes [7]. In the next step, we performed univariate testing of each variable for the three binary outcomes (number of complications, presence of a major complication and length of hospital stay) using the chi-square test. In case multiple variables showed a significant association, we performed multivariate logistic regression analysis for the respective outcome measurements. For those variables that showed a statistically significant association with outcomes in the multivariable analysis, we used the Mann-Whitney U test to compare the groups for favorable and unfavorable outcomes. All statistical tests were performed using the Statistical Package for the Social Sciences 23.0 software (SPSS Inc., Chicago, IL, USA). All tests were two-sided with the significance level set at *p* < 0.05.

## 3. Results

The preoperative, intraoperative and postoperative characteristics of the 317 patients who underwent RC are presented in Table 1. One third of patients (107, 34%) received a continent urinary diversion. The distribution of patients into three groups according to preoperative characteristics, vital signs and blood values is presented in Table A1. Preoperative potassium levels were classified as either low (≤4.28 mmol/L, 33% of patients), average (4.29–4.67 mmol/L, 33% of patients) or high (≥4.68 mmol/L, 33% of patients).

Patients were divided into favorable and unfavorable outcome groups for each of the three outcomes: number of complications (low vs. high), presence of a major complication (no vs. yes), and length of hospital stay (shorter vs. longer). The favorable outcome groups comprised 186 patients (58%) with ≤2 total complications, 209 patients (66%) without a major complication (Clavien-Dindo grade complications ≤2) and 150 patients (47%) with a hospital stay of ≤21 days. Accordingly, the unfavorable outcome groups comprised 131 patients (42%) with ≥ 3 total complications, 108 patients (34%) with major complications (Clavien-Dindo grade complications ≥3) and 167 patients (53%) with a hospital stay of ≥22 days.

The results of the univariate analysis for associations between preoperative factors and postoperative complications and the length of hospital stay are presented in Table 2. A low Charlson comorbidity score (*p* = 0.011), low ASA score (*p* = 0.015), no aspirin intake (*p* = 0.048) and high-normal preoperative potassium level (*p* = 0.034) were identified as preoperative factors associated with the favorable outcome of ≤2 postoperative complications. High-normal preoperative potassium was the only preoperative factor associated with a shorter hospital stay of ≤21 days (*p* = 0.007).

Table 3 shows the results of the multivariate logistic regression analysis. Among the four preoperative factors associated with the favorable outcome of ≤2 postoperative complications (low Charlson comorbidity score, low ASA score, no aspirin intake and high-normal preoperative potassium level), only high-normal preoperative potassium remained an independent predictor for ≤2 postoperative complications (odds ratio 0.67, 95% confidence interval (0.48, 0.92), *p* = 0.014).

In line with these findings, preoperative potassium was significantly higher in patients with ≤2 postoperative complications (4.53 ± 0.45 mmol/L) than in those with ≥3 postoperative complications (4.42 ± 0.42 mmol/L; *p* = 0.032), and in patients with a hospital stay of ≤21 days (4.58 ± 0.45 mmol/L) compared to those with a hospital stay of ≥22 days (4.39 ± 0.43 mmol/L; *p* < 0.001), but not in those with no major complications (4.53 ± 0.57 mmol/L) compared to those with major complications (4.43 ± 0.43 mmol/L; *p* = 0.062). The effect of low-normal, average-normal and high-normal preoperative potassium levels on the number of postoperative complications per patient and length of hospital stay are shown in Table 4.

## 4. Discussion

We performed a retrospective study on 317 consecutive patients who underwent RC to investigate the association of preoperative variables with clinical (complications) and economical (length of hospital stay) postoperative outcomes. We found that high-normal preoperative potassium levels were associated with fewer complications and a shorter hospital stay for patients undergoing RC.

Our findings fill an important knowledge gap in the literature about the association of preoperative variables with postoperative complications after RC. While perioperative management factors have been well reviewed, such as the link between blood transfusion and adverse oncological outcomes [7], the role of electrolyte levels in patients undergoing RC has not been investigated.

In general, the balance of electrolytes and water seems to play an important role in the recovery of gastrointestinal function after abdominal surgery. In this regard, a positive balance of 3 kg delayed the normalization of gastrointestinal function and prolonged the length of hospital stay in patients undergoing elective colon surgery [8]. A positive balance for in-hospital patients undergoing major surgery is usually due to increased perioperative fluid substitution. As these fluids contain either average or low levels of potassium (especially in the case of glucose infusion), substitution might lead to lower blood potassium levels. The problem of low postoperative potassium levels might be further enhanced by reduced nutrition intake and the use of diuretics in the early postoperative period. Although fast-track concepts have been established, nutrition intake is reduced after major surgery. The human body receives potassium from daily nutrition intake; however, a reduced intake is expected in the early postoperative period after RC. Moreover, the use of loop diuretics to reduce the aforementioned positive balance might further contribute to low postoperative potassium levels if not accompanied by adequate potassium substitution. With regard to these factors, it is possible that the higher potassium levels observed in our study led to fewer complications and earlier discharge by achieving faster activation of gastrointestinal function.

Potassium, an osmotically active cation, plays a key role in the formation of nerve action potentials and, consequently, the regulation of smooth muscle neuronal activity. As a result, changes in potassium balance pose a great threat to patients, especially after urinary diversion. In general surgery, the P-Possum score is used to predict the prognosis of morbidity and mortality after surgery. One of the many factors contributing to this score is the potassium level at the time of surgery. Lu and colleagues reported that potassium levels below 3.5 mmol/L increase the P-Possum score, and, therefore, the likelihood of a negative outcome after major surgery [9].

The effect of electrolyte levels on clinical outcomes is not limited to patients undergoing abdominal surgery. Investigations of levels of sodium and potassium in patients undergoing coronary artery bypass surgery showed that electrolyte abnormalities led to a significantly longer stay in the intermediate care unit [10]. This suggests that potassium levels are not only crucial for postoperative gastrointestinal activation after abdominal surgery, but also have an influence on systemic recovery after surgery in general. The type of urinary diversion (continent vs. incontinent) had no impact on length of hospital stay in our study, whereas an association between continent ileal neobladder and longer hospital stay had been reported before [11].

Our study is the first to investigate the influence of preoperative potassium levels on postoperative complications and length of hospital stay in patients undergoing RC and urinary diversion. We acknowledge that one inherent limitation of our study is its retrospective design. Moreover, supplementation of potassium in the postoperative period was not standardized, and was usually only undertaken when blood potassium levels dropped below the normal value. Despite these limitations, a major strength of our study is the identification of a simple intervention that could potentially lead to improved outcomes. A randomized controlled trial should be performed to investigate whether there is a causal relationship between preoperative potassium supplementation and postoperative complications and length of hospital stay.

## 5. Conclusions

In our retrospective study, a high-normal preoperative potassium level was associated with lower 30-day morbidity and a shorter hospital stay in patients undergoing RC. We recommend that a randomized controlled trial be performed to investigate whether there is a causal association between preoperative potassium supplementation and postoperative complications and length of hospital stay.

## Figures and Tables

**Table 1 jcm-11-01174-t001:** Preoperative, intraoperative and postoperative characteristics of the 317 patients who underwent radical cystectomy.

Variable (Unit)	Mean ± SD/*n* (%)
**Preoperative patient characteristics**	
Age (years)	69 ± 11
Male gender	238 (78%)
BMI (kg/m^2^)	25.8 ± 5.3
Charlson comorbidity score	3 ± 2.1
ASA classification	3 ± 0.6
Aspirin intake	65 (23%)
Polypharmacy (>5 drugs)	79 (29%)
**Preoperative vital signs**	
Systolic blood pressure (mm Hg)	130 ± 18
Diastolic blood pressure (mm Hg)	80 ± 10
Heart rate	76 ± 12
**Preoperative blood values**	
Preoperative hemoglobin	12.9 ± 2.7
Preoperative leucocytes	7.9 ± 3.7
Preoperative thrombocytes	261 ± 108
Preoperative C-reactive protein	0.6 ± 3.5
Preoperative creatinine (mg/dL)	1.1 ± 0.6
Preoperative MDRD	66 ± 23
Preoperative glucose	98 ± 35
Preoperative sodium (mmol/L)	141 ± 3.1
Preoperative potassium (mmol/L)	4.5 ± 0.4
**Intraoperative characteristics**	
Estimated blood loss (mL)	1300 ± 1151
Operative time (min)	399 ± 97
Continent urinary diversion	107 (34%)
Incontinent urinary diversion	210 (66%)
**Postoperative outcome parameters**	
**Histopathology**	
T0	25 (8%)
T1	55 (19%)
T2	76 (26%)
T3	99 (33%)
T4	42 (14%)
Positive lymph nodes	81 (26%)
Positive resection margin	38 (12%)
**Complications according to the Clavien-Dindo classification**	
Patients with any complication	258 (81%)
Highest complication grade I	25 (8%)
Highest complication grade II	125 (39%)
Highest complication grade IIIa	38 (12%)
Highest complication grade IIIb	23 (7%)
Highest complication grade IVa	32 (10%)
Highest complication grade IVb	7 (2%)
Highest complication grade V	8 (3%)
Number of complications per patient	2 ± 2.2
**Length of hospital stay (days)**	22 ± 13

ASA = American Society of Anethesiologists Score; BMI = Body Mass Index; MDRD = Modification of Diet in Renal Disease.

**Table 2 jcm-11-01174-t002:** Univariate analysis of the association between preoperative factors and postoperative outcome measurements. *p*-values are presented. Statistically significant results are marked in bold.

Preoperative Factor	Outcome Measurements (*p*-Values Shown)		
	≥3 Postoperative Complications	Major Complication (Clavien ≥ 3)	Hospital Stay ≥22 days
Age	0.526	0.310	0.950
BMI	1.000	0.621	0.779
**Charlson comorbidity score**	**0.011**	0.286	0.138
**ASA classification**	**0.015**	0.717	0.220
**Aspirin intake**	**0.048**	0.605	0.956
Polypharmacy (>5 drugs)	0.276	0.078	0.746
Systolic blood pressure	0.174	0.075	0.188
Diastolic blood pressure	0.313	0.240	0.647
Heart rate	0.180	0.881	0.892
Urinary diversion	0.405	0.473	0.053
Preoperative hemoglobin	0.092	0.921	0.445
Preoperative leucocytes	0.469	0.406	0.355
Preoperative thrombocytes	0.886	0.875	0.203
Preoperative CRP	0.123	0.136	0.073
Preoperative creatinine	0.057	0.449	0.158
Preoperative MDRD	0.309	0.479	0.919
Preoperative glucose	0.531	0.084	0.294
Preoperative sodium	0.236	0.950	0.391
**Preoperative potassium**	**0.034**	0.109	**0.007**

ASA = American Society of Anethesiologists Score; BMI = Body Mass Index; MDRD = Modification of Diet in Renal Disease; CRP = C-reactive protein.

**Table 3 jcm-11-01174-t003:** Multivariate logistic regression analysis to identify preoperative factors that predict the presence of three or more complications after radical cystectomy. Statistically significant results are marked in bold.

Preoperative Factor	Odds Ratio	95% Confidence Interval	*p* Value
Charlson comorbidity score	1.29	0.93, 1.78	0.128
ASA score	1.67	0.98, 2.85	0.058
Aspirin intake	1.72	0.93, 3.17	0.083
**Preoperative potassium**	**0.67**	**0.49, 0.92**	**0.014**

**Table 4 jcm-11-01174-t004:** Effect of preoperative potassium level on the number of postoperative complications per patient and length of hospital stay.

	Preoperative Potassium Level		
Postoperative Outcome	Low-Normal (≤4.28 mmol/L), *n* = 106	Average-Normal (4.29–4.67 mmol/L), *n* = 106	High-Normal (≥4.68 mmol/L), *n* = 105
Number of complications per patient, mean ± SD	2.7 ± 2.3	2.6 ± 2.5	2.4 ± 1.9
Length of hospital stay in days, mean ± SD	28 ± 13	24 ± 12	23 ± 11

## Data Availability

The data that support the findings of this study are available on request from all authors up on request. The data are not publicly available due to restrictions e.g., their containing information that could compromise the privacy of research participants.

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
