# Peer review of "High–Normal Preoperative Potassium Level Is Associated with Reduced 30–Day Morbidity and Shorter Hospital Stay after Radical Cystectomy"

_jcm, 2022, doi:10.3390/jcm11051174_

Round 1

Reviewer 1 Report

The authors investigated the association between preoperative potassium level and clinical (30-day morbidity) and economical (length of hospital stay) postoperative outcomes of patients undergoing radical cystectomy. They retrospectively evaluated clinical data of 317 patients who had undergone radical cystectomy for bladder cancer. They found that high preoperative potassium level was associated with better clinical (lower 30-day morbidity) and economical (shorter hospital stay) outcomes in patients undergoing radical cystectomy.

An interesting study with simple hypothesis and rationale. Enjoyed the study. Several questions still need to be addressed:

  1. Are the surgeries in this study performed by a single surgeon?
  2. Did these patients receive neo-bladder or ileal conduit surgery? Did they use colon, intestine or stomach as bladder substitution? These might have a role in blood electrolytes and should be evaluated.

Reviewer 2 Report

The authors evaluated the impact of the level of preoperative potassium levels on the complications and hospital stay of patients undergoing RC. the manuscript is well written however major limitations should be addressed.

1- the variables included in univariable and multivariable analyses should include the type of diversion as previous studies have shown that type of diversion can impact hospital stay (for example https://doi.org/10.1016/j.clgc.2018.01.004)

2-The authors should provide justification for the selected three subcategories of potassium levels, and should mention of any of the patients were defined as hyper or hypokalemic (normal potassium 3.5-5.2).  If all patients are within the normal range of potassium level,  the names of the subgroups should be low normal, average-normal and high normal potassium.

Furthermore, classification into three groups with a small number of patients can overestimate the effect size. Thus, classifying the patients into only two groups of low-normal and high-normal will be more appropriate unless the current classification was justified by previous literature. 

3- The authors should use the pre-defined cutoffs of potassium levels in the univariable and multivariable analyses rather than continuous   variable

Furthermore, the authors should write between brackets next to each variable in the univariable and multivariable analyses whether this variable continuous or dichotomous variable

for example 

preoperative hemoglobin (continuous)

or 

preoperative hemoglobin (high vs. low)

Round 2

Reviewer 1 Report

No further comments.

Reviewer 2 Report

The authors have addressed all comments and the manuscript now should be acceptable as is.